# *n*-Alkanes and *n*-Alkenes in Virgin Olive Oil from Calabria (South Italy): The Effects of Cultivar and Harvest Date

**DOI:** 10.3390/foods10020290

**Published:** 2021-02-01

**Authors:** Angelo Maria Giuffrè

**Affiliations:** Dipartimento di Agricoltura, Università degli Studi Mediterranea di Reggio Calabria, Contrada Melissari, 89124 Reggio Calabria, Italy; amgiuffre@unirc.it

**Keywords:** aliphatic hydrocarbon, virgin olive oil, human health, linear hydrocarbon, minor component, unsaponifiable

## Abstract

*n*-Alkanes and n-alkenes are components of the unsaponifiable fraction of an olive oil. These were analysed by GC on-column analysis and are here proposed as an additional tool to certify the origin, authenticity, traceability and chemical quality of olive oil produced in the Reggio Calabria province (South Italy). Nine cultivars were studied: Cassanese, Coratina, Itrana, Leccino, Nociara, Ottobratica, Pendolino, Picholine and Sinopolese grown in the region of Calabria (South Italy). *n*-Alkanes in the range from 21 to 35 chain carbon atoms and alkenes in the range from 23:1 to 25:1 chain carbon atoms were found with the following elution order: heneicosane (C21), docosane (C22), tricosene (C23:1), tricosane (C23), tetracosene (C24:1), tetracosane (C24), pentacosene (C25:1), pentacosane (C25), hexacosane (C26), eptacosane (C27), octacosane (C28), nonacosane (C29), triacontane (C30), entriacontane (C31), dotriacontane (C32), tritriacontane (C33), tetratriacontane (C34), pentatriacontane (C35). The oil of all cultivars showed a decreasing trend in total *n*-alkane and *n*-alkene content, with the oil of Sinopolese showing the highest content, varying from 328.50 to 214.00 mg/kg. Odd-chain alkanes predominated over even-chain *n*-alkanes, and tricosane, tetracosane and pentacosane were the most represented alkanes. Cultivar and harvest date significantly influenced the *n*-alkane and *n*-alkene content. These findings can be useful to distinguish different olive cultivars and to decide the fruit harvest date for the oil of the Reggio Calabria province (South Italy). A daily quantity of 30 g of olive oil of the Sinoplese cv (the one with the highest *n*-alkane and *n*-alkene content) was found to be in accordance with the suggestions of the European Agency for the evaluation of medicinal products Committee for veterinary medicinal products and biogenic hydrocarbons intake for the human diet.

## 1. Introduction

Olive oil is globally recognised for its biological properties due to its physico-chemical composition. Calabria (South Italy) is the second largest Italian region for olive oil production (755,032 tons in 2016) [1], and for this reason many works have been conducted to study olive oil produced in this geographic region with regard to the physico-chemical properties of the fruit and oil of many olive cultivars [2,3,4,5,6,7,8,9,10,11], and in particular with regard to the evolution during olive fruit ripening of triglycerides [12] and minor components such as sterols [13], fatty alcohols [14] and waxes [15].

The unsaponifiable fraction also contains linear hydrocarbons, among them alkanes and alkenes, which have not been much studied, probably because both European regulation [16] and the Trade Standard of the International Olive Council [17] do not yet consider these components.

*n*-Alkane content was found to be affected by the geographic origin of the oil [18] and can be used to detect enhanced concentrations of leaf material in olive oil, because *n*-alkane profiles of olive leaves were found not to exhibit significant variations during the harvest season in contrast to *n*-alkanes in olive oil [19].

Studies conducted on vegetable oils found that *n*-alkane composition varies depending on the botanical origin [20]; for instance, the carbon atom chain of *n*-alkane ranges between 21 and 34 mg/kg in avocado pulp oil [21] and between 19 and 32 mg/kg in tomato seed oil [22].

To study the hydrocarbon fraction, it is useful to compare the endogenous hydrocarbons (alkanes and alkenes) with the exogenous hydrocarbons (mineral oils and paraffins) [23,24,25]. Alkanes are metabolized to fatty alcohols and then acids in the small intestine [26] and in the liver [27]. Mineral oils and paraffins are retained in human tissues [28] and can migrate, for example from paper board to dry foods or from jute or sisal bags [29,30].

Endogenous hydrocarbons are produced by the decarboxylation of long-chain fatty acids [31,32]; they are different in aquatic or terrestrial plants and are related to the geographic origin.

Exogenous hydrocarbons are generally due to non-voluntary (albeit dangerous for human health) addition mainly caused by contamination due to air pollution [33], incorrect storage [34], or incorrect processing [35] and transport.

From the point of view of the edibility of a vegetable oil, it is important to point out that the intake of biogenic hydrocarbons is normally within the range of 10–100 mg/person/day (0.17–1.7 mg/kg of body weight/day), and the intake of hydrocarbons by all sources has been estimated at around 240 mg/person/day (4 mg/kg body weight/day), if a 60 kg person is considered, as suggested by the European Agency for the evaluation of medicinal products [36].

*n*-Alkanes have also been used as markers in nutritional studies with wild ruminant and non-ruminant animals [37,38].

In a recent work, we studied the influence of harvest season on the *n*-alkane and *n*-alkene content in virgin olive oils produced in the region of Calabria (South Italy) [39] in a scientific context of scarce information with regard to the occurrence of *n*-alkane content in vegetable oils and, in particular, of olive oils [40].

This is the first complete work studying the evolution of endogenous alkanes and alkenes during fruit ripening in monocultivar olive oil and studying the effect of cultivar and harvest date on the alkane and alkene composition, in order to characterise monocultivar olive oil and to have a further decisional element in relation to the date of olive harvest.

## 2. Materials and Methods

### 2.1. Plant Material

All olive trees were grown in mono cultivar plantations, in a non-polluted geographic area of Rizziconi, in the Gioia Tauro Plan at 115 m on the sea level (Calabria, South Italy).

Nine cultivars were selected for this work: two autochthonous (Ottobratica and Sinopolese) and seven allochthonous (Cassanese, Coratina, Itrana, Leccino, Nociara, Pendolino and Picholine) for this geographic area. The authentication of these cultivars was conducted by the author of this study (an agronomist) in collaboration with the owner of the olive grove and the supplier of the plants. The ground is flat, alluvial, with silt and sand. The climate is humid and temperate and olive trees were not irrigated. The maximum rain fell was 165 mm in March 2016 and the minimum was 2 mm in June 2017.

The maximum temperature of this geographic area was 39.4 °C on August 2017 (in the morning) and the minimum was −3.0 °C on January 2017 (in the night). Thirty plants aged 25 to 35 years, healthy and uniform in size, grown along a line between two opposite corners of the orchard, were chosen for each cultivar. Pruning was conducted every two years, but dead wood was removed each year. All cultivars were own-rooted plants, and the same fertilisation criterion was applied each year (N, P and K in a ratio 20/10/10). The main parasitic attacks were due to: *Bactrocera oleae*, *Spilocaea oleaginea* and *Colletotricum gloeosporioides* and treatments were applied to contrast these pathogens.

The experiment was conducted for two harvest years (2016–2017 and 2017–2018) on the following harvest dates: 3 October, 18 October, 3 November, 17 November, 5 December, 19 December, 3 January. Sampling was manually and carefully conducted at biweekly intervals, and from each cultivar, 2 kg olives/tree (for a total of 60 kg) were randomly picked at each harvest date until drupes were found on trees. The maturity index (MI) varied from 0 (100% intense green colour of skin and pulp) to 7 (skin and pulp 100% black) [41]. The MI of the last sampling was 6 for fruits of cultivars whose last sampling was conducted on 5 and 19 December, and it was 7 for fruits picked on January. This was related to the different response given from an olive genotype to a specific environment and it is the reason for what we have conducted five, six or seven samplings in relation to the different cultivars. In Coratina, Itrana, Leccino and Picholine cultivars, no fruit was found on plants on 19 December. Drupes were processed within 5 h after picking in a small mill “Mini 30” (AGRIMEC Valpesana, Calzaiolo, S. Casciano VP, Florence), with a capacity of 40 kg with the following 10-step procedure: (1) drupe separation by leaves, stems and any solid material; (2) mild drupe washing; (3) drupe crushing by a hammer-mill at room temperature; (4) mixing of the olive paste without adding water for 35 min at a range of temperatures between 18 and 20 °C; (5) positioning of the olive paste in a pile of circular steel grids; (6) pressure by a hydraulic press with a continuous slow and mild increase in pressure up to 200 bar; (7) maintaining maximum pressure for 20 min; (8) separation of the olive oil from the liquid mixture (oil and waste water) by a laboratory centrifuge (3000 rpm for 10 min); (9) filtration of the oil by filter paper; (10) storage of the olive oil in 100 mL amber glass bottles in the dark at 15–20 °C until analysis, which was conducted within two days after oil extraction.

### 2.2. Chemicals

Chemicals (analytical grade and chromatographic grade) were from Carlo Erba, Milan, Italy. Pure standards of alkanes were from Sigma Chemical Co., St Louis, MO, USA. Silica gel was from (Merck, Darmstadt, Germany).

### 2.3. Analytical Procedure

The European Regulation [6] and the Trade Standard of the International Olive Council [7] do not contain a specific method for alkanes and alkenes determination; however, Annex XVII of the European regulation (Method for the determination of stigmastadienes in vegetable oils) [6] can be well applied to determine also alkanes and alkenes in olive oil. In brief, *n*-hexacosane diluted in n-hexane as internal standard was placed in a 250 mL glass flask and n-hexane was evaporated by a mild stream of nitrogen. At this point, olive oil and alcoholic potash were added to 10%, then the reflux condenser was started, and the mixture was heated to slight boiling for 30 min until complete saponification reaction. The unsaponifiable fraction was separated and then was carefully introduced in a separating glass column containing a previously activated silica gel and *n*-hexane. At this point, the chromatographic elution was started with *n*-hexane at a flow rate of 1 mL/min approximately. The first 35 mL of eluate contains *n*-alkanes and *n*-alkenes.

### 2.4. Gas Chromatography

For gas-chromatographic analysis of *n*-alkanes and *n*-alkenes, a Carlo Erba HRGC3000 instrument was used, which was equipped with an on-column injector, an FID detector and a capillary column SE-54 MEGA-Milano-Italy (column length 25 m, ID 0.32 mm and film thickness 0.25 μm). The oven temperature was programmed at 60 °C for 1 min and then at 5 °C/min up to 315 °C (40 min). The detector temperature was maintained at 330 °C. Peaks were identified by comparing their retention indices with those of pure standards of *n*-alkanes and *n*-alkenes.

### 2.5. Statistical Analysis

Two batches of 30 kg drupes each were prepared for each cultivar at each harvest date, and two independent replicates were conducted from the oil of each batch, with a total of four replicates/cultivar/harvest date. Excel 2010 software was used to calculate the means and standard deviations of eight replicates (four replicates/year × two harvest years). The means were analysed by one-way ANOVA and Tukey’s test, at 5% probability, using the SPSS 17.0 software (SPSS Inc., Chicago, IL, USA); the variables were the cultivar and the harvest date. Principal component analysis (PCA) was applied by SPSS software for Windows, version 15.0 (Chicago, IL, USA).

## 3. Results and Discussion

Alkanes in the range of C21 to C35 and alkenes in the range of C 23:1 and C 25:1 were detected with the following elution order: heneicosane (C21), docosane (C22), tricosene (C23:1), tricosane (C23), tetracosene (C24:1), tetracosane (C24), pentacosene (C25:1), pentacosane (C25), hexacosane (C26), eptacosane (C27), octacosane (C28), nonacosane (C29), triacontane (C30), entriacontane (C31), dotriacontane (C32), tritriacontane (C33), tetratriacontane (C34), pentatriacontane (C35).

C21 was found almost always in a quantity lower than 1 mg/kg, except in the last two samplings of Picholine and in all samples of Sinopolese. The cultivars produced very high significant differences among the means (*p* = 0.000). The differences between harvest dates were: *p* = 0.009 for Nociara, *p* = 0.003 for Sinopolese, *p* = 0.002 for Itrana, *p* = 0.001 for Leccino, Ottobratica and Pendolino, *p* = 0.000 for all other cultivars (Appendix A). C21 was revealed to be the first odd-chain *n*-alkane in rangeland plant species from the Sudan, with a quantity ranging between 1 and 8 mg/kg DM of leaves or whole plant [42].

The lowest C22 value was found in the fifth sampling of Itrana (0.14 mg/kg), whereas the very highly significant highest value (*p* = 0.000) was in the fourth sampling of Sinopolese (16.93 mg/kg) which showed a different alkane profile with respect to the other studied cultivars (Appendix A). Sayago et al. [43] studied seventeen Spanish olive oils of different cultivars and found 4 mg/kg as C22 mean content (1.35 mg/kg min–9.93 mg/kg max) in lower quantity than our Sinopolese oil and in higher quantity with respect to all other cultivars. C22 was found to be less than 1% in the aerial parts of three *Artemisia* species in which a range between C19 and C33 was detected [44].

In Appendix A, data regarding to C23:1 are listed, which is the first *n*-alkene appearing in the GC profile, and which almost always amounted to less than 1 mg/kg, with the exception of the first sampling of Sinopolese (1.24 mg/kg). Nociara showed no significant differences in C23:1 during fruit ripening (*p* = 0.061) (Appendix A). The C23:1 content varied, showing very highly significant differences with olive ripening (*p* = 0.001) in Itrana and (*p* = 0.000) in all other cultivars, and it was similarly influenced (*p* = 0.000) by cultivar at each harvest date (Appendix A). Osorio-Bueno et al. studied the oil of seven olive cultivars and found a C23:1 content ranging between 0.02 and 0.11 ppm [45].

C23 showed a tendency in decrease in the oil of all cultivars from the first part of the harvest season to the last sampling. Cassanese always very highly significantly showed the lowest C23 content (*p* = 0.000), ranging from 1.05 to 0.42 mg/kg, whereas Sinopolese oil very highly significantly showed the highest values (*p* = 0.000), ranging from 90.48 to 60.03 mg/kg (Appendix A). Sakoui et al. [46], in Maski oil of complete mature fruits, found a C23 content of 10.4 mg/kg, in lower quantity than C23 found in Picholine, Sinopolese and in the initial harvest season of Nociara and Ottobratica oils but in higher quantity than in all other cultivars and samplings.

The second detected *n*-alkene was C24:1, accounting for less than 1 mg/kg, except in the first sampling of Ottobratica and Sinopolese (1.05 and 1.17 mg/kg, respectively). This hydrocarbon showed a decreasing trend in many cultivars with Ottobratica, Picholine and Sinopolese, very highly significantly accounting for the highest contents (*p* = 0.000) in all samplings. Leccino and Pendolino showed no significant differences during fruit ripening (*p* = 0.249 and 0.296 respectively) (Appendix A). It has to be pointed out that the C24:1 content of the oil of almost all the allochthonous cultivars was similar to that found by other authors in Spanish olive oils (0.02–0.09 mg/kg) [43].

C24 was characterised by a very highly significant decreasing trend during olive ripening in all cultivars (*p* = 0.000). Leccino, Ottobratica, Picholine and Sinopolese oils accounted for more than 10 mg/kg in the first sampling and Picholine and Sinopolese oils were above this threshold during the whole harvesting season. Both cultivar and harvest date variables showed very highly significant differences (*p* = 0.000) between the C24 values (Appendix A). Osorio Bueno et al. [45] studied the hydrocarbon content of virgin olive oil from the Extremadura region (Spain) and found a C24 content ranging between 0.53 and 1.62 ppm, i.e., in a lower amount in many cases with respect to our findings.

C25:1 was the last eluted of the three detected alkenes. It showed a decreasing trend in eight out of the nine studied cultivars; in fact, only Itrana oil showed an increasing (but nonlinear) trend. The harvest date influenced differently the C25:1 content in the oil of all the studied cultivars: Cassanese, Itrana, Nociara, Ottobratica, Picholine and Sinopolese (*p* = 0.000); Coratina and Pendolino (*p* = 0.010 and *p* = 0.003, respectively); Leccino (*p* = 0.009), (Appendix A). Koprivnjak and Conte studied the oil of Leccino, Bianchera, Carbonazza and Busa cultivars grown in the Pula area (Croatia) and calculated a C25:1 content ranging between 0.15 mg/kg (Busa cv) and 1.23 mg/kg (Bianchera cv) [47], similar to our findings (Appendix A).

C25 was highest in Ottobratica Picholine and Sinopolese oils (*p* = 0.000). Ottobratica oil showed a very highly significant decrease with ripening (*p* = 0.000) from 34.15 to 11.40 mg/kg. The oil of Picholine ranged from 27.95 mg/kg to 21.05 mg/kg between samplings (*p* = 0.000), and the oil of Sinopolese varied from 91.35 to 59.63 mg/kg (*p* = 0.000). Cultivar showed very highly significant differences (*p* = 0.000) in the C25 content at each sampling (Appendix A). Sakoui et al. [46] found C25 as the major alkane in Maski oil of complete mature fruits, accounting for 22.1 mg/kg, in a higher quantity than in the oil of all samplings of six out nine cultivars of our work, whereas in a study on pulverized leaves of *Kalanchoe pinnata*, C25 was quantified in traces and as the first *n*-alkane eluted in the GC chromatogram [48].

C26 was very highly significantly found to have the highest quantity (*p* = 0.000) in Ottobratica oil (4.52–2.53 mg/kg) and in Sinopolese oil (8.75–5.15 mg/kg), namely the two autochthonous cultivars. Both cultivar and harvest date showed very highly significant differences (*p* = 0.000) between the contents of this saturated hydrocarbon during sampling (Appendix A). Sayago et al., in 70 olive oils of cultivars grown in southwest Spain—Beas, Gibraleón, Niebla, Sanlúcar de Guardiana, Abequina, Picual, and Verdial de Huévar—found a mean C26 content of 0.32 mg/kg (0.10 mg/kg minimum–1.09 mg/kg maximum) [43]. This is explained by the high content of short-chain *n*-alkanes in these Spanish oils and the low content of the long-chained, confirmed by Lazon et al. [49] and in contrast to the results of our work on the olive oils of Reggio Calabria province (south Italy) (Appendix A).

C27 content showed no significant differences (*p* = 0.233) in Picholine during olive ripening, but decreased in all other cultivars. The variation was significant (*p* = 0.022) in Itrana from 3 October (5.43 mg/kg) to 5 December (3.66 mg/kg) and was very highly significant (*p* = 0.000) in the remaining seven cultivars (Appendix A). Koprivnjak et al. [50] analysed by LC-GC the oil of three olive cultivars grown in the geographic area of Istria (Croatia) and found a decreasing tendency in the content of each alkane even if with a different rate; in addition, in the oil of Leccino and Buza cultivars, the major compounds were C23, C24 and C25 (similarly to our cultivars), whereas in Bjelica oil, the major compounds were (C25, C27 and C29). Bianchi et al. found C27 homologue to be the main *n*-alkane (47% of the total) in black olives of Coratina cv grown on hills surrounding Pescara (central Italy) [51]. The findings of Mihailova et al. showed that in olive oils produced in Slovenia, central Italy, Portugal northern Greece and Spain, C27, C29 and C31 *n*-alkanes were prevalent [18], but we also found this in the olive oils of the west area of the Reggio Calabria province, south Italy (Appendix A).

C28 was highest in Coratina, Ottobratica, Pendolino and Sinopolese and a very highly significant (*p* = 0.000) variation in diminution during olive ripening was shown in all cultivars. The highest C27 value was in the first sampling of Pendolino (4.05 mg/kg), and the lowest was in the fifth and last sampling of Itrana (0.53 mg/kg) (Appendix A).

C29 content was more than 20 mg/kg in the first sampling of Ottobratica, Pendolino and Sinopolese oils and in the second sampling of Pendolino and was not less than 3.27 mg/kg in any of the samplings of all cultivars (Appendix A). In Spanish virgin olive oil from the Extremadura region, C29 was found to be one of the predominant alkanes in a range of 2.17–7.02 ppm [45]. C29 was the *n*-alkane detected in highest quantity (27% of the total *n*-alkane content) in green olives of Coratina cv from central Italy [51] and in highest quantity in seed oil of sunflower (49.63%), sesame (18.45%), peanut (12.69%) and safflower (27.05%) [52]. At the same time, C29 was the main *n*-alkane (24.4%), and the last one detected in the GC profile of *Hypericum perforatum* dried leaves purchased from Norsk Medisinaldepot, Oslo, Norway [53].

C30 showed very highly significant differences (*p* = 0.000) when cultivar and harvest dates were compared. Pendolino had the highest content in the first two samplings (2.88 and 2.48 mg/kg respectively). Itrana had the lowest content in all samplings varying from 0.38 mg/kg to 0.25 mg/kg), (Appendix A).

C31 content never exceeded 10 mg/kg and was very significantly the highest (*p* = 0.000) in Pendolino, ranging from 9.63 mg/kg (3 October) to 4.05 mg/kg (19 December). There was a depleting trend, even if the content of the last sampling of six out nine cultivars was higher than the last but one (Appendix A). In a study conducted in Tuscany (central Italy) on Frantoio, Moraiolo and Leccino cultivars, C31 was found to be the highest *n*-alkane in olive leaves, and was present in higher quantities in olive leaves than in olive fruits; this condition was repeated each month in samplings from July to November [19].

C32 accounted for less than 1 mg/kg in all samplings of all cultivars with the exception of the content of the fourth sampling conducted for Coratina (1.27 mg/kg). The harvest date did not influence the C27 content of Ottobratica (*p* = 0.055), whereas very highly significant differences were found in all other cultivars (*p* = 0.000). When considering the cultivar effect, we found highly significant differences (*p* = 0.009) on 19 December and very highly significant differences (*p* = 0.000) in all other harvest dates (Appendix A). Wilson et al. (1999) studied pregnant sows (*Sus scrofa* L.) and measured faecal recovery of *n*-alkanes from ingested grass and from dosed *n*-alkane (C32) under controlled indoor conditions, with or without the addition of soybean oil in the diet. They found no effect of soybean (*Glycine max* L.) oil addition on faecal recovery of alkanes, nor was there a consistent pattern of diurnal variation in faecal *n*-alkane concentration. Faecal recoveries were incomplete and ranged from 43.3% for C24 to 98.6% for C32 [38].

C33 was within the range of 3.02 mg/kg at the first sampling of Sinopolese oil and 0.27 mg/kg of Pendolino oil at the fourth sampling. When considering all harvest dates, Itrana and Picholine contained the lowest C33 values, which were always observed at values less than 0.74 mg/kg. When the cultivar variable was considered, highly significant (*p* = 0.002) differences were found between the values of the sixth sampling, and very highly significant (*p* = 0.000) differences were observed between all other samplings (Appendix A). C33 was the highest *n*-alkane revealed (62.90%) on pulverized leaves of *Kalanchoe pinnata* [48].

C34 was one of the alkanes detected in the lowest quantity—at values always less than 0.27 mg/kg—in Ottobratica on 19 December. Highly significant differences (*p* = 0.006) were found when comparing the means of the sixth sampling, and very highly significant differences (*p* = 0.000) were found when comparing all other samplings. Itrana and Picholine oils showed the lowest C34 values, i.e., less than 0.06 mg/kg (Appendix A). Mihailova et al. found C34 in higher quantities in olive leaves than in olive fruits throughout the whole ripening season; this happened also for C30, C32, C33 and C35. In brief, this condition was found for all the high-chain *n*-alkanes and, consequently, the oil extracted from batches of olive fruits containing a high leaf content showed a high *n*-alkane content [19].

C35 was the last alkane eluted in the chromatogram. For each cultivar, the highest value was found in the first sampling, and ranged between 0.69 mg/kg (Sinopolese) and 0.12 mg/kg (Picholine). The differences between means were significant (*p* = 0.015) in Itrana, highly significant in Coratina (*p* = 0.004), and very highly significant in the oil of all other cultivars (*p* = 0.000) (Appendix A).

The total *n*-alkane content was influenced both by the harvest date and by the cultivar effects (*p* = 0.000). This was particularly evident for Ottobratica and Sinopolese (the two autochthonous cultivars), whose total alkane content was much higher than other ones. When considering the first sampling, the oil of Sinopolese contained 324.76 mg/kg, i.e., 10.68 times more than in Itrana (30.40 mg/kg) and 9.29 times more than in Cassanese (34.97 mg/kg). When considering the fifth sampling (on 5 December, when fruits were found on the trees of all cultivars), the oil of Sinopolese contained 266.98 mg/kg of total alkanes, with higher content with respect other cultivars as follows: 19.72% more than in Cassanese, 8.44% more than in Coratina, 16.82% more than in Itrana, 8.92 more than in Leccino, 8.00% more than in Nociara, 3.78% more than in Ottobratica, 7.13% more than in Pendolino, and 4.03% more than in Picholine (Appendix A). El Antari et al. [54] studied the total *n*-alkane content of five olive oils from Morocco and found the following values: 19.65, 27.94, 55.97, 101.63 and 42.73 mg/kg in oils obtained from different cultivars growing in different geographic areas and from olives picked with different ripening indexes. Mihailova et al. [19] studied the ripening effect of the alkane content and found a depletion with ripening in fruits, while simultaneously observing stability in leaves; consequently, the oil extracted from fruits processed with a high leaf content showed a higher *n*-alkane content with respect to oil produced with a low leaf content. Pineda et al. [55] studied the total alkane content in Hojiblanca extracted oil submitted to four successive malaxations with a small-scale extractive plant and found that the *n*-alkane content increased constantly with the number of malaxations from 16.50 to 25.00 mg/kg. In the same study, Pineda et al. found a decreasing *n*-alkane content in olive fruit of the Picual cultivar until July and a substantial stabilisation from July to December [55]. Similar results were found by Sakouhi et al. in oil of Meski cv grown in north-east Tunisia, with a decreasing trend from the 21st to the 26th week after flowering and a substantial stabilisation from the 27th week [46].

Gómez-Coca et al. studied the effect of the number of extractions and the n-alkane content in the oil of three olive cultivars (Aloreña, Leccino and Manzanilla) grown in the south of Spain, and found that in the olive oil of the second centrifugation, the *n*-alkane content was higher than in the oil of the first one. At the same time, the oil obtained with solvent from olive pomace contained more *n*-alkanes that the centrifuged one [56]. *n*-Alkane content of endogenous origin can be used to distinguish extra virgin olive oil or lampante olive oil from olive pomace oil; in fact, Moret et al. found that olive oils obtained using a mechanical system (pressure or first centrifugation) usually do not contain mineral paraffins above the detection limit (1 mg/kg), whereas solvent-extracted pomace olive oils showed a high mineral paraffin content. This could be associated with the transport of olive pomace from the olive mill plant to the refining plant [23]. Paraffins, and in general, mineral oil saturated hydrocarbons, are undeliverable, because exposure to them in rat diet was found to increase liver and spleen weights, as well as the vacuolization and granuloma formation associated with lymphoid cell clusters in the liver [57]. With regard to the *n*-alkane content in different plants and the possibility of distinguishing them with chemotaxonomic distinctions, it should be pointed out that *n*-alkane production appears to be much higher in angiosperms than in gymnosperms [58].

The harvest date produced highly significant differences in the total *n*-alkene content of Cassanese, Coratina and Nociara oils (*p* = 0.002, *p* = 0.003 and *p* = 0.002 respectively) and very highly significant differences (*p* = 0.001 and *p* = 0.000) in the oil of all other cultivars. A constant decreasing tendency was observed in alkene content even if, in some cases, the last sampling showed a slightly higher content than the previous one. The cultivar variable always influenced the total alkene content (*p* = 0.000). No total alkene content was found to be as high as 3.75 mg/kg (found in Sinopolese oil at the first sampling). More specifically, in the oils of Cassanese, Itrana, Leccino, Nociara and Pendolino, the total alkene content was lower than 0.39 mg/kg. On the fifth sampling (5 December), six cultivars produced an oil with a total alkene content ≤0.35 mg/kg, whereas Ottobratica, Picholine and Sinoplese oils exhibited values of 0.90, 1.67 and 2.03 mg/kg, respectively (Appendix A). The literature on *n*-alkane composition in olive oil is scarce, but that on *n*-alkenes is extremely poor. In one of the existing papers, the authors studied the oil of Leccino, Bianchera, Carbonazza and Busa cultivars grown in the Pula area (Croatia) and found a total *n*-alkene content ranging between 0.26 mg/kg (Busa cv) and 3.88 mg/kg (Carbonazza cv) [47].

The sum of alkanes and alkene contents was consistent with the behaviour of each single hydrocarbon fraction. The decreasing trend was confirmed, and the harvest date and cultivar effects caused very highly significant differences between means (*p* = 0.000). The diminution with ripening from the first to the last sampling in the total alkanes and alkenes content was: 173% in Cassanese (from 35.17 to 20.33 mg/kg); 208% in Coratina (66.50–32.00 mg/kg); 191% in Itrana (30.50–16.00 mg/kg); 252% Leccino (75.50–30.00 mg/kg); 167% Nociara (58.50–35.00), 245% Ottobratica (139.50–57.00 mg/kg); 222% Pendolino (82.00–37.00 mg/kg); 135% Picholine (91.50–68.00 mg/kg); 154% Sinopolese (from 328.50 to 214 mg/kg), (Figure 1). The total *n*-alkane and *n*-alkene content studied by other authors with respect to the Leccino oil produced in the Pula area (Croatia), allochthonous for this geographic area, was 61.67 mg/kg, even though the harvesting date and the maturation index of the olive fruits were not specified in this study; Bianchera, Carbonazza and Busa cultivars possessed 47.81, 47.05 and 40.15 mg/kg, respectively [47]. If assuming a daily use of 30 g olive oil in the human diet, it can be stated that 9.86 to 6.42 mg/day (Sinopolese oil), 4.19 to 1.71 mg/day (Ottobratica oil), 1.06 to 0.61 mg/day (Cassanese oil) or 0.92 to 0.48 mg/day (Itrana oil) is ingested in the diet, that is, less than the 10–100 mg/person/day of biogenic hydrocarbons and the 240 mg/person/day of hydrocarbons stated by all sources indicated by the European Agency for the evaluation of medicinal products [36].

During ripening, the total alkanes/total alkenes ratio was not affected by the harvest date in the oil of Pendolino (*p* = 0.148), while it showed significant differences in Coratina, Nociara and Picholine oils (*p* = 0.023, *p* = 0.021 and *p* = 0.012, respectively), highly significant differences in Sinopolese oil (*p* = 0.008), and very highly significant differences in Itrana and Ottobratica oils (*p* = 0.000). The cultivar effect produced very highly significant differences at each harvest date (*p* = 0.000). The three highest values were found in Pendolino: 926.37, 959.96 and 693.34 in the third, fourth and fifth samplings, respectively. The three lowest values were found in the first three samplings of Picholine: 35.58, 31.92 and 36.64 (Appendix A).

The values related to the sum of the total odd-chain alkanes were predominant in the oils of the two autochthonous cultivars: Sinopolese with a decreasing tendency from 241.70 mg/kg to 160.18 mg/kg, and Ottobratica with a decreasing tendency from 107.60 to 46.31 mg/kg. Itrana oil contained the lowest total of even carbon chain number alkanes, from 23.78 to 12.90 mg/kg (decreasing with drupes ripening), i.e., 10.16 to 15.14 lower than in Sinopolese oil (Appendix A). Bortolomeazzi et al. [59] studied the total odd chain number alkanes in the oil of some Italian cultivar and found 52.1 and 53.7 ppm in Leccino cv grown in the Abruzzo region (central Italy), and 37.4, 43.2 and 33.8 ppm in oil of Coratina of the Apulia region (south Italy), that is, almost the same content we found in the oil obtained in the first sampling (3 October) of Leccino (56.32 mg/kg), and a similar content to our second and fourth samplings of Coratina (46.95 and 32.45 mg/kg) (Appendix A). Odd-chain *n*-alkanes were found to be useful for distinguishing different genus and species of oils. Specifically, C31 was found to be the most abundant in peanut oil; C31 was predominant in grapeseed oil, followed by C29 and C27; C29 and 31 were prevalent in sunflower and corn oils; and C29 was prevalent in hazelnut oil, followed by C31 and C27 [20]. Partially different results were found by Benitez-Sanchez et al. in crude hazelnut oils from different geographic areas, whereby C31 was found in low quantities to trace amounts [60].

Additionally, the values related to the sum of the total even-chain alkanes were greater in the two autochthonous cultivars. In Sinopolese oil, the rate during fruit ripening was decreased from 83.06 to 52.10 mg/kg (159% less), and in Ottobratica, the rate decreased from 28.47 to 9.61 mg/kg (296% less). Itrana oil showed the lowest absolute values, as well as a depletion of 223% (from 6.62 to 2.97 mg/kg from the first to the last sampling) (Appendix A). The findings of Bortolomeazzi et al. [59] describe a total even-chain alkanes of 10.1 and 10.2 ppm for Leccino oil (central Italy) and of 8.7, 9.2 and 7.0 ppm for Coratina oil (South Italy), with a similar content to our Leccino at the end of October and our Coratina oil at the end of November (Appendix A).

The total odd/even chain number alkanes ratio is described in Figure 2. The harvest date did not influence this value in Itrana, while it showed significant differences in Leccino oil (*p* = 0.017), highly significant differences in the oil of Pendolino (*p* = 0.005), and very highly significant differences (*p* = 0.000) in all other cultivars. When considering the cultivar effect, the differences between means were always very highly significant (*p* = 0.000). This ratio ranged between 2.73 (in Picholine oil on 18 October) and 5.17 (in Ottobratica oil on 5 December).

Principal component analysis (PCA) is a multivariate statistical analysis method used to reduce the number of variables, and this was applied to identify which compounds provided the best ability to differentiate the cultivars. The analysis of data using the PCA technique made it possible to chemically group the samples into three groups, in such a way as to express and evidence their similarities and differences. The significant factor loadings from the PCA of chemical constituent variables were obtained as suggested by D’Agostino et al. [61] after varimax rotation. The eigenvalues of the covariance matrix showed that the set of the two principal components (PCs) accounted for 86.14% of the total variance in the dataset with respect to cultivars. The loadings of first and second principal components (PC1 and PC2) accounted for 54.16 and 31.98% of the variance, respectively (Figure 3). Positive values for PC1 indicate the cultivars with the highest contents of C21, C22, C23:1, C23, C24:1, C24, C25:1, C25, C26, ∑ (total) Alkanes + ∑ (total) Alkenes, ∑ (total) alkenes, ∑ (total) even chain alkanes, and ∑ (total) odd chain alkanes. The highest PC2 values correspond to the cultivars with C28, C29, C30, C31, C32, C33, C34, C35. Cultivar projection on the factorial map allows a clear separation between the analysed samples (Figure 1). In fact, PCA analyses indicate the existence of three groups. The first group was formed of Leccino, Cassanese, Itrana, Nociara and Picholine cultivars, the second of Coratina, Ottobratica and Pendolino cultivars and third grouped of Sinopolese cultivar. According to the bi-dimensional representation of the first two factors, Sinopolese oil has a profile far removed from the other cultivars. Pendolino, Ottobratica and Coratina oils are more correlated with PC2 than the Sinopolese oil. This means that the Sinopolese cultivar is very different from the other cultivars.

## 4. Conclusions

A marked significant difference between cultivars was found with Sinopolese and Ottobratica oils containing the highest alkanes and alkenes content; in contrast, Cassanese and Itrana oils significantly showed the lowest content at each sampling. Both alkane and alkene content constantly and significantly decreased with fruit ripening in the oil of all cultivars. The significant differences found in this work showed that alkanes and alkenes can be used as an additional tool for certifying the origin, authenticity, and traceability of the oil of the studied cultivars grown in the Reggio Calabria province (South Italy). With regard to the daily intake, it can be confirmed that if a daily use of 30 g olive oil is considered in the human diet, it can be assumed that 6.42 to 9.86 mg/day (Sinopolese oil), 1.71 to 4.19 mg/day (Ottobratica oil), 0.61 to 1.06 mg/day (Cassanese oil) and 0.48 to 0.92 mg/day (Itrana oil) are ingested in the diet, i.e., amounts well within the 10–100 mg/person/day of biogenic hydrocarbons and the 240 mg/person/day of hydrocarbons by all sources indicated by the European Agency for the evaluation of medicinal products.

## Figures and Tables

**Figure 1 foods-10-00290-f001:**
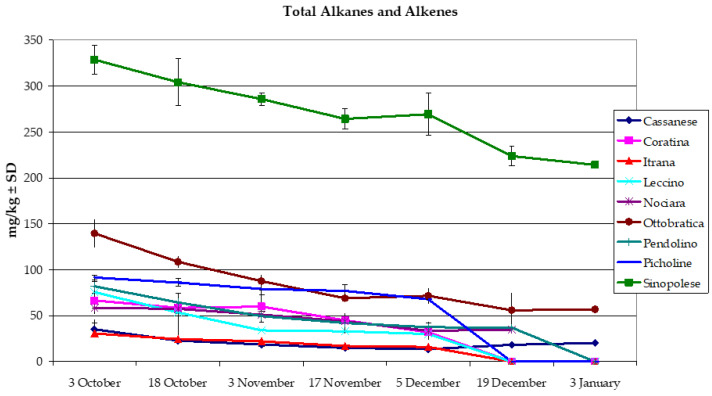
Total alkanes and alkenes. Data are expressed as mg/kg and are given as mean (*n* = 8). Means and standard deviations were calculated on 8 replicates (4 replicates/year x two harvest years), in the harvest years 2016–2017 and 2017–2018.

**Figure 2 foods-10-00290-f002:**
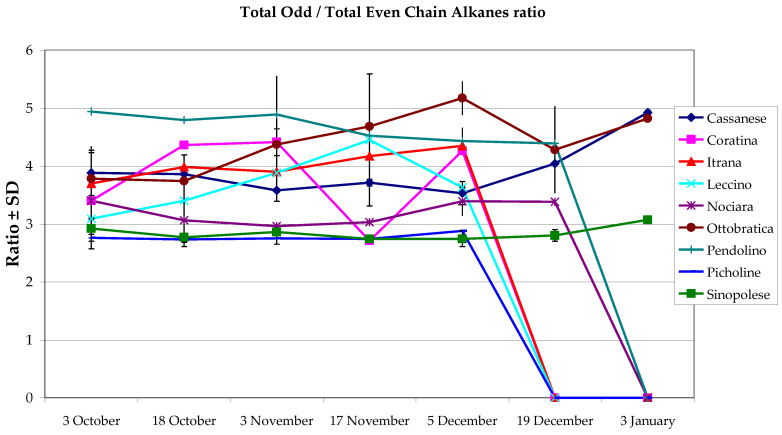
Odd-/even-chain alkanes ratio. Data are given as means (*n* = 8). Means and standard deviations were calculated on eight replicates (four replicates/year x two harvest years) in the harvest years 2016–2017 and 2017–2018.

**Figure 3 foods-10-00290-f003:**
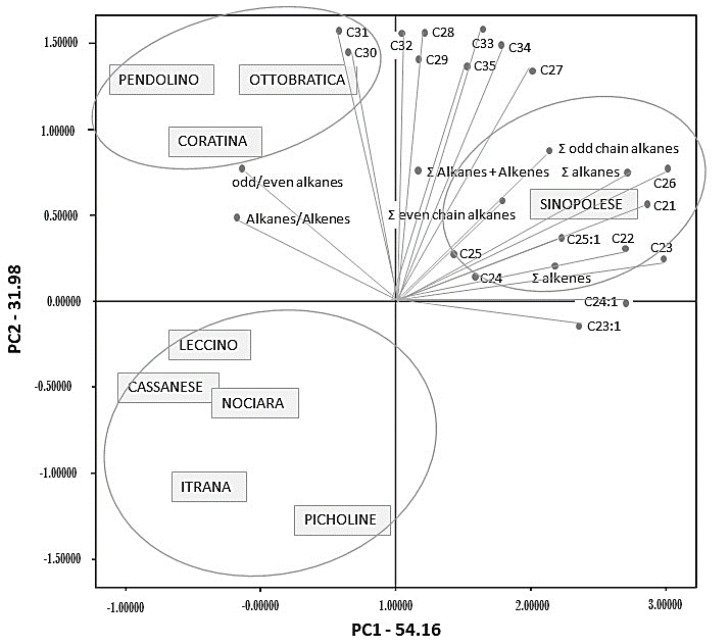
Biplot graph, PC1 versus PC2, using loadings and scores for different olive oil cultivars.

## Data Availability

Not applicable.

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
