# Peer review of "n-Alkanes and n-Alkenes in Virgin Olive Oil from Calabria (South Italy): The Effects of Cultivar and Harvest Date"

_foods, 2021, doi:10.3390/foods10020290_

Round 1

Reviewer 1 Report

3. Results and discussion - at the beginning  there is a chapter that need to be erased (left from  template version)

 - too many tables, try to reduce tables and present part of results in the form of diagram instead

  • some tables can be presented in section Supplementary Materials
  • literature needs to be  adjusted according  journal proposition
  • in the text, when you talk about Table 1, it should be stated as Table 1, not Tbl.1 ; do not use abbreviations- also look carefuly once again guidlines for the authors
  • Tables should be placed in the main text near to the first time they are cited.
  • English language and style need to be  checked  and minor corrections are needed
  • please add some other bibliographic references (other authors) with research of Calabrian olive oils 

Author Response

Reply to the Reviewer
I really want to thank you for your comments. I have included (in blue color) all corrections you have suggested. Thank you for your valuable remarks and time!

3. Results and discussion - at the beginning there is a chapter that need to be erased (left from template version)
Thank you. Chapter deleted;

-too many tables, try to reduce tables and present part of results in the form of diagram instead

The manuscript has been revised accordingly. 12 tables were moved to the supplementary material section and 2 tables were changed as figures;

-some tables can be presented in section Supplementary Materials

The manuscript has been revised accordingly;

-literature needs to be adjusted according journal proposition

Thank you. Literature adjusted;

-in the text, when you talk about Table 1, it should be stated as Table 1, not Tbl.1 ; do not use abbreviations- also look carefuly once again guidlines for the authors

Thank you. Correction done;

-Tables should be placed in the main text near to the first time they are cited.
Thank you. Tables are now placed in the main text near to the first time they are cited;

-English language and style need to be checked and minor corrections are needed Thank you for this comment. English improved;
-please add some other bibliographic references (other authors) with research of Calabrian olive oils

Thank you fro your comments. Ten recent references were added with regard to the olive oil produced in the Calabria Region and 32 authors more were cited

Reviewer 2 Report

I received a manuscript with visible corrections and additions, probably after the first review. The manuscript contains important issues concern for the presence of n-alkanes and n-alkenes, constituents of the unsaponifiable fraction of olive oil. The Author took into account the guidelines of the European Agency for the evaluation of medicinal products, Committee for veterinary medicinal products and biogenic hydrocarbons intake for the human diet. It was shown that the variety and date of harvest significantly influenced the content of n-alkanes and n-alkanes, which may be useful for assessing the use of different olive varieties and deciding on the date of harvesting. As shown by the Author, this is the first work to investigate the evolution of endogenous hydrocarbons during fruit ripening in single-grade olive oil. A very large research scope has been presented. The results can be very useful in practice, mainly for the selection of varieties and the degree of ripeness of the olives. However, there can always be other factors related to the climate, weather.

Tables: “Means values in a vertical column with different capital letters and means in a line 238 with small letters were significantly different according to the Tukey test (*, p≤ 0.05; **, p≤  0.01; ***, p≤  0.001).” - Was the use of three p-values needed? This information is missing in the methodology. Moreover, in each table, only p at the level of 0.001 gave significantly different values. I have doubts whether the statistical analysis was performed correctly or the same asterisks were copied everywhere. I think it is better to give specific values of all p-values in each table for both factors instead of asterisks. Correction of statistical analysis must be corrected throughout the manuscript!

The manuscript prepared properly, well written and organized. The scope of research is appropriate to the topic of the work. The discussion of the obtained results is noteworthy. These references correctly applied. Many older and recent years sources were used. In general I have no comments to the title, keywords, introduction, materials and methods, experimental procedures, conclusions, references.

The presented statistical analysis is the biggest problem. I hope the Author will correct it.

Some other comments:

Lines 22-23, 600, 601… : “ the oil of Sinopolese showing the highest content varying 22 from 328.50 to 214 mg/kg.” - Write the range of values ​​starting with the smaller ones and use the same precision (number of decimal digits).

Lines131-132: “Peaks were identified by comparing their retention  indices with those of pure standards of n-alkanes and n-alkenes and with literature data” - Examples of sources should be cited.

Lines 591-593: “Alkane gas-chromatographic profile showed a characteristic bell shape with tricosane, tetracosane and pentacosane as the predominant ones when compared with all other alkanes and alkenes across

all cultivars.” Are these 2 initial sentences needed in conclusions? No information about this in the manuscript.

Figures: Figures should be corrected, there is no complete description of the axis and standard deviations, which is very important in these studies. In the captions of tables and figures, there is no information which data relates to the range of 2016-2017 and 2017-2018.

Author Response

Ø Response to the Reviewer 2

Dear Reviewer, thank you for your suggestion to improve my work. I have written in red the corrections that you suggested.

  • the statistic was corrected both in the tables and in the text as you suggested;
  • Lines 22-23, 600, 601… : “ the oil of Sinopolese showing the highest content varying 22 from 328.50 to 214 mg/kg.” - Write the range of values starting with the smaller ones and use the same precision (number of decimal digits).
  1. Lines 22-23 the decimal digits are now included after 214.
  2. In the conclusions section, the range of values starts with the smaller ones.
  • 4 section. The text was corrected as you suggested, in fact no literature data I have found for retention indices of alkanes and alkenes;
  • The first sentence of the conclusions section was deleted;
  • Figures 1 and 2. The standard deviation was included and the axis description was improved;
  • The range of harvest years was included in tables and figures;
  • Lines 22-23, 600, 601… : “ the oil of Sinopolese showing the highest content varying 22 from 328.50 to 214 mg/kg.” - Write the range of values starting with the smaller ones and use the same precision (number of decimal digits).
  1. Lines 22-23 the decimal digits are now included after 214.
  2. In the conclusions section, the range of values starts with the smaller ones.
  • 4 section. The text was corrected as you suggested, in fact no literature data I have found for retention indices of alkanes and alkenes;
  • The first sentence of the conclusions section was deleted;
  • The standard deviation was included and the axis description was improved;
  • The range of harvest years was included in tables and figures.

Reviewer 3 Report

The paper entitled  " n-Alkanes and n-alkenes in virgin olive oil from Calabria (South Italy): the effects of cultivar and harvest date" sent to the journal  Foods, has a lot of things to improve. The authors have done a lot of experimental work and as a result have a high amount of data. However, to get the information they have it will be necessary to use more than ANOVA. I suggest that they employ some unsupervised data processing techniques, such as Principal Component Analysis (PCA). With the data you have and using this chemometric technique you can surely apply your methodology to authenticate olive oils. Commenting on the composition tables one by one does not serve this purpose. The evolution of these compounds over time is already known, since, as the authors themselves indicate, there is a degradation of the long chain fatty acids.

For what said before the work needs a great revision since the treatment of data must be improved substantially, and to pass the tables of composition of all the compounds to the complementary material. 

Author Response

Ø Response to the Reviewer 3

Dear Reviewer, thank you for your suggestion to improve my work. I have written in blue the corrections that you suggested.

  • The PCA was conducted and one figure and discussion were included;
  • All tables were moved to the supplementary material section.

Round 2

Reviewer 2 Report

I recommend to improve the accuracy of the probability data, you should use the same accuracy in the decimal notation, i.e. instead of "p = 0", it should be "p = 0.000", some of this data is written Italic, others not correct it.

I also recommend checking the whole manuscript, there are commas before numerical citation, also at the end of the sentence instead of dots.

Also need to correct the standard deviations on the figures, they only have one arm?

Author Response

Ø Response to the Reviewer 2

Dear Reviewer, thank you for your suggestion to improve my work. I have written in red the corrections that you suggested.

 I recommend to improve the accuracy of the probability data, you should use the same accuracy in the decimal notation, i.e. instead of "p = 0", it should be "p = 0.000", some of this data is written Italic, others not correct it.

  1. R. Corrections done in the text and in the tables.

I also recommend checking the whole manuscript, there are commas before numerical citation, also at the end of the sentence instead of dots.

  1. R. Corrections done.

Also need to correct the standard deviations on the figures, they only have one arm?

  1. R. Correction done, two arms.

Reviewer 3 Report

The article has been satisfactorily modified, so I have no further comments.

Author Response

Thank you for your comments